# Concentration changes of atmospheric F-gases and analysis of their potential sources at Zhongshan Station, Antarctica, 2021

Ruiqi Nan<sup>1,2#</sup>, Biao Tian<sup>1#\*</sup>, Xinfeng Ling<sup>3</sup>, Weijun Sun<sup>2</sup>, Yixi Zhao<sup>1</sup>, Dongqi Zhang<sup>1</sup>, Chuanjin Li<sup>4</sup>, Xin Wang<sup>1</sup>, Jie Tang<sup>1</sup>, Bo Yao<sup>5,6\*</sup>, Minghu Ding<sup>1,7</sup>

- 5 ¹State Key Laboratory of Disaster Weather Science and Technology, Chinese Academy of Meteorological Sciences, Beijing, 100081, China
  - <sup>2</sup>College of Geography and Environment, Shandong Normal University, Jinan, 250014, China
  - <sup>3</sup>Shouxian Meteorological Bureau, Huainan, 232000, China
  - <sup>4</sup>Polar Research Institute of China, Shanghai, 200136, China
- Operation of Atmospheric and Oceanic Sciences & Institute of Atmospheric Sciences, Fudan University, Shanghai, 200438, China
  - <sup>6</sup>Meteorological Observation Centre of China Meteorological Administration, Beijing, 100081, China
  - <sup>7</sup>Key Laboratory of Polar Atmosphere-ocean-ice System for Weather and Climate, Ministry of Education of the People's Republic of China, Shanghai, 200438, China
- 15 #These authors contributed equally to this work and should be considered co-first authors.

Correspondence to: Biao Tian (tianbiao727@foxmail.com), Bo Yao (yaobo@fudan.edu.cn)

Abstract. As potent greenhouse gases with high global warming potentials, fluorinated gases (F-gases) have emerged as significant contributors to global radiative forcing. Owing to minimal anthropogenic influences, Antarctica provides an exceptional natural environment for investigating background atmospheric F-gas concentrations. This study presents the first comprehensive report of temporal variations in 11 F-gas species at the Zhongshan National Atmospheric Background Station (ZOS; 69.4° S, 76.4° E) throughout 2021. This study is the first to provide concentration changes of 11 F-gases at ZOS in Antarctica in 2021. The datasets are publicly available at the National Tibetan Plateau Data Center at <a href="https://doi.org/10.11888/Atmos.tpdc.302283">https://doi.org/10.11888/Atmos.tpdc.302283</a> (Tian et al., 2025). The concentrations of most F-gases significantly increased throughout 2021 at ZOS. The concentrations of F-gases in East Antarctica were greater than those in the Antarctic Peninsula and the interior on the basis of data comparisons with three other Antarctic stations. Back trajectory and clustering analyses using the HYSPLIT model revealed that the contributions of different trajectory clusters were nearly identical at each station. Source apportionment analysis via the PMF model identified industrial processes, refrigeration, fire suppression, and electronics as key contributors to F-gas concentrations in the Antarctic atmosphere. While the one-year observation period precludes long-term trend assessment, these high-frequency measurements capture the baseline variability critical for detecting future anomalies. Continuous multiyear monitoring at ZOS is necessary to establish statistically robust growth rates.

## 1 Introduction

60

Fluorinated greenhouse gases (F-gases), which contain fluorine as the sole halogen, such as hydrofluorocarbons (HFCs), perfluorocarbons (PFCs), nitrogen trifluoride (NF<sub>3</sub>), and sulfur hexafluoride (SF<sub>6</sub>), have gained attention due to their significant global warming potential (GWP). Unlike ozone-depleting substances (ODSs) which are known for their role in ozone layer depletion, F-gases exert no direct impact on stratospheric ozone, resulting in a zero ozone depletion potential (ODP) (Supplementary Materials Table S1). F-gases, which are primarily synthetic, have emerged as the main substitutes for ODSs after their phase-out under the Montreal Protocol. However, their growing concentrations have raised concerns about their role in global warming, leading to the inclusion of HFCs under the Kigali Amendment in 2016 to limit their production and consumption globally.

F-gases primarily comprise HFCs, PFCs, NF<sub>3</sub>, and SF<sub>6</sub>, nearly all of which are synthetic in nature. The primary sink for HFCs is their reactions with hydroxyl radicals (OH), whereas photolysis and reactions with Cl or O (<sup>1</sup>D) radical are considered minor sinks (Kurylo et al., 2003; Vollmer et al., 2011; Vladimir L. Orkin., et al. 2020; Thompson et al., 2024).

The lifetimes of the primary HFCs vary significantly, ranging from less than 1.5 years to as long as 242 years. For example, PFC-116, CF<sub>4</sub>, and NF<sub>3</sub> exhibit lifetimes of 10,000, 50,000, and 550 years, respectively, as indicated in Supplementary Materials Table S1. If released into the atmosphere, these substances can accumulate, and their effects can persist for centuries or millennia. Moreover, these substances can be transmitted to the stratosphere and the rest of the world.

Since 1977, the NOAA/ESRL has analyzed minor atmospheric constituents sampled at 12 remote stations (Montzka et al., 2015; Takeda et al., 2021). The Advanced Global Atmospheric Gases Experiment (AGAGE) and its precursors have been pivotal in measuring and analyzing the composition of the global atmosphere, providing critical data for understanding climate change and environmental monitoring since 1978. AGAGE is distinguished by its ability to measure all important species identified in the MP and the major non-CO<sub>2</sub> greenhouse gases at the global scale, at a high frequency, and at multiple stations. These substances exhibit atmospheric abundance levels ranging from the ppm (parts per million, i.e., one in 106) scale down to the ppq (parts per quadrillion, i.e., one in 1015) scale. Global monitoring revealed that under the control of the MP, the Kyoto Protocol, and the United Nations Framework Convention on Climate Change (UNFCCC), the CFC concentration decreased, the growth rate of HCFCs initially decreased and then stabilized, and the HFC concentration continued to increase (Yi, L. et al., 2021, 2023).

In Antarctica the isolated polar vortex results in some of the cleanest air on Earth, providing an ideal location for studying background concentrations of atmospheric gases. Although there have been limited F-gas measurements in this region(Takeda et al., 2021), recent studies have begun to fill this gap. NOAA operates only two stations monitoring F-gas in Antarctica, NILU operates only one station in Antarctica. Moreover, all the Antarctic stations have not yet achieved continuous automatic observations of F-gases. Existing research has focused on air trapped in ice cores, firn air, and archived air. Ongoing measurements of select ODSs and HFCs were initiated by NOAA via flasks at 7 sampling locations only from

the mid- to late 2000s (Montzka, et al. 2015). Early research aimed to analyze firn air samples from Law Dome, Antarctica, 65 for several ODSs listed in the MP, encompassing virtually the entire history of anthropogenic emissions of CFCs, HCFCs, and halons and providing early 20th-century levels of CH<sub>3</sub>CCl<sub>3</sub> and CCl<sub>4</sub> (Sturrock et al., 2002). The estimated global mixing ratios, trends, and emissions derived from the analysis of archived air samples exhibit high verified uncertainties (Montzka, et al. 2015). In addition to direct observation, the atmospheric abundance of HFC-23 was retrieved with a groundbased Fourier transform infrared (FTIR) spectrometer, and the spectra observed at Syowa Station, Antarctica (69° S, 39.6° E) were analyzed (Takeda et al., 2021). However, the results of inversion also need to be verified by in situ observational data. In this study, atmospheric samples were collected via a stainless steel SUMMA tank at the Zhongshan National Atmospheric Background Station (ZOS) in Antarctica in 2021 and analyzed through the ODS-5pro system via gas chromatography-mass spectrometry (GC-MS), and concentration data of 11 types of F-gases were obtained. These data were compared with data from three Antarctic stations of NOAA and NILU to analyze F-gases concentration differences. The data were also compared with data from the midlatitude Cape Grim Observatory in the Southern Hemisphere and the Zeppelin station in the New Orson region to determine the global concentration of F-gases at ZOS. A backward air mass trajectory model was used to calculate backward trajectories of gas masses and analyze the possible sources and transmission paths of F-gases at ZOS. This research fills the gap in China's halogenated hydrocarbon research in Antarctica, provide a data basis for understanding the concentration level, transport and sedimentation mechanisms, emission inversion study of F-gases, and help to assess the effectiveness of international protocols.

## 2 Station and data description

## 2.1 Data acquisition process at ZOS

#### 2.1.1 Sample collection

ZOS (69.4° S, 76.4° E, 18.5 m above sea level (asl)), which was established in 1989, is China's second Antarctic scientific research station. Located at the coastal margin of Las Man Hills in Prydz Bay, the station occurs near the Antarctic continental ice sheet. An atmospheric composition observatory occupies a level terrain west of Swan Ridge at the northwestern perimeter of the station. Katabatic winds develop as cold air masses descend through topographic depressions, generating sustained downslope flows. Meteorological records indicate year-round easterly dominance, with mean wind speeds exceeding 7 m s<sup>-1</sup>. The air masses influencing the station originate primarily from Antarctic continental coastal zones and northern oceanic sectors. This strategic location ensures minimal anthropogenic influences coupled with sufficient atmospheric mixing. Peer-reviewed studies have validated the ability of the station to capture representative atmospheric background concentrations (Wang, C. et al., 2016; Ding, M. et al., 2020; Tian, B. et al., 2023).

In 2021, we implemented a biweekly ambient air sampling regimen at ZOS. The sample container encompassed stainless steel electropolished canisters 3 L in volume (LabCommerce, Inc. Santa Clara, CA, U.S.A.). The interior of each canister was subjected to inertification, and air samples can be stored for a long time without adsorption. The sampling protocol comprised three sequential phases: (1) The sampler is connected to the canister through stainless steel bellows, and system integrity is verified through pressurization to 10 psi with 1 min of leak monitoring; (2) a 10-min purge is applied to eliminate residual contaminants in the canister; and (3) the canister is charged to 30 psi with ambient air, after which it is sealed and stored properly. A total of 26 ambient air samples were collected (Supplementary Materials Figure S1). A comprehensive metadata registry (Table S2) provides critical parameters, including ambient temperature, barometric pressure, and meteorological observations, to ensure analytical traceability.

### 2.1.2 Analysis of the air samples

95

100

The canisters carrying air samples were transported back to China by icebreaker Xue Long during the 39th Chinese National

Antarctic Research Expedition. The concentrations of 11 F-gases (Supplementary Materials Table S1) in the samples were
analyzed via the ODS5-pro system. This system comprises five core subsystems: an automatic sampling module, an analysis
system, a standard gas measurement system, an auxiliary gas (helium and nitrogen) measurement system, and a data
processing system. The analysis system encompasses a refrigerant-free preconcentration module, a GC instrument (A91P,
Panna Instruments Co., Ltd., Jiangsu, China), and a quadrupole MS detector (7700B, Suzhou Anyeep Instrument Co., Ltd.,
Jiangsu, China). Ambient air was dried with a two-stage Nafion dryer and preconcentrated in a stainless-steel trap (1.60 mm
ID, packed with HayeSep). Major background gases (N2, O2, Ar, Xe, CH4, CO2) were purged with high-purity helium under
precise temperature control. The retained compounds were transferred to a second trap (0.51 mm ID, HayeSep) for
reconcentration, then thermally desorbed, separated by GC (GasPro precolumn, PoraBOND Q analytical column), and
detected by quadrupole MS. The GC temperature and pressure programs followed the Medusa GC–MS protocol(Miller, B. R.
et al., 2008), with a total analysis time of 70 minutes.

The quantitative method for concentation of each F-gas in the background atmosphere is the same as that commonly employed by the international observation networks of AGAGE and NOAA, namely, a bracket air sample (A) with a standard gas sample (S) (S-A-S-A-S) is used to correct the detection drift during long-term observations. Because the chromatographic peak area (or peak height) responds linearly to the mole fraction of a given compound at its background level, the quantitative method for determining mole fractions of halogenated gases in the air sample  $C_A(ppt)$  is as follows:

$$C_{A} = \frac{2 \times C_{s} \times A_{A}}{A_{S1} + A_{S2}} \quad \text{or} \quad C_{A} = \frac{2 \times C_{s} \times H_{A}}{H_{S1} + H_{S2}}$$
 (1)

where  $C_A$  is the known mole fraction (ppt) of the target compound in standard gas;  $A_A$  and  $H_A$  are the chromatographic peak area and peak height, respectively, of the compound in the air sample (dimensionless); and  $A_{S1}$  and  $A_{S2}$  ( $H_{S1}$  and  $H_{S2}$ ) denote

the chromatographic peak areas (peak heights) of the compound in two standard gas samples before and after air sample measurement, respectively.

The measurement accuracy and precisions of ODS5-pro system is estimated by 38 times of replicate measurements of the same standard gas. The detection limit of the ODS5-pro system was determined using a signal-to-noise (S/N) method(Yi et al., 2023). Accuracy is defined by the mean relative error and precision by the relative standard deviation. Each air sample measurement was bracketed by a reference gas (working standard) measurement to detect and correct for drift in the detector sensitivity. The working standard gases used at measure instrument were compressed ambient air stored in high-pressure cylinders and calibrated against quaternary standards, which were calibrated by a tertiary standard from the Scripps Institution of Oceanography (SIO). Our measurements are reported on the derived calibration scale. The F-gases studied in this paper correspond to three types of SIO, namely: SIO-05 (CF4, HFC-134a, HFC-152a), SIO-07 (PFC-116, HFC-23, HFC-143a), SIO-12 (NF3), SIO-14 (HFC-125, HFC-227ea, HFC-236fa).

The measurement precision of ODS5-pro system depends on the mole fractions of compounds. The precisions are around 0.5%, 0.5%–1%, 1%–4%, and 4%–11%, for the species with mole fractions greater than 100, 20–100, 2–20, and 0.1–2 ppt, respectively. Intercomparison studies have demonstrated a measurement consistency between AGAGE Medusa-GC/MS systems and ODS5-pro system within ±2% (Yi et al., 2023). Comprehensive system configuration details have been reported in Yi et al. (2023), with operational schematics and performance metrics provided in Figure S2 and Table S3 During the test, one stainless steel canister exhibited air leakage, and the data efficiency reached 96.2%. The validated dataset has been archived at the National Tibetan Plateau Data Center (https://doi.org/10.11888/Atmos.tpdc.302283).

#### 2.2 Observations from other stations

In this study, we retrieved additional Antarctic measurements from two fully intercalibrated affiliated measurement programs. Details of these programs are summarized in Table 1 and Figure 1. The global terrain model uses ETOPO2v2 developed by the National Geophysical Data Center (NGDC) affiliated with NOAA. Specifically, observational data were obtained from the NOAA network (https://gml.NOAA.gov/aftp/data; last access: 11 April 2025) and the NILU network (https://ebas-data.nilu.no/Default.aspx; last access: 11 April 2025). GC – MS was employed at all stations for analysis, ensuring consistency and comparability across measurements (Montzka et al., 1996, 2015; Montzka and Dutton, 2012; Platt et al., 2024; Lunder, 2024).

Figure 1: Spatial distributions of the four F-gas monitoring stations in the Antarctic.

Palmer Station, Antarctica (PSA; 64.8° S, 64.1° W; 15 m asl), is located on Anvers Island, just outside the Antarctic Circle. At this station, air samples are collected in glass flasks at a biweekly frequency. Trollhaugen Station, Antarctica (TRL; 72° S, 2.5° E; 1,553 m asl), operated by NILU, is located approximately 235 km inland from the coast in Jutulsessen, Dronning Maud Land. At Trollhaugen Station, ambient air samples are collected in stainless steel flasks at a biweekly frequency. In addition to these coastal stations, we collected air samples at the South Pole Observatory (SPO; 90° S, 24.8° W, 2,837 m asl), which represents atmospheric conditions over the Antarctic interior. At SPO, paired-flask sampling is conducted at weekly to biweekly frequencies (Montzka et al., 2015; Ashwin Mahesh, 2003).

In addition to data from Antarctic stations, we collected data at other latitudes. Midlatitude baseline measurements were obtained from the Cape Grim Observatory (CGO, 40.7° S, 144.7° E, 164 m asl) in Tasmania, Australia, which is a global baseline station. The station is located at the northwestern corner of Tasmania and is subject to mainly westerly winds, rendering it ideal for sampling unpolluted air, as the sampled air arrives at Cape Grim after traveling long distances over the Southern Ocean, under conditions described as baseline conditions (Crawford, et al., 2017). Those samples were shipped back to the Boulder laboratory, where they were analyzed via at least 3 GC systems. At the Zeppelin Station (ZEP, 78.9° N, 11.9° W, 474 m asl), operated by NILU, observations are made in situ using GC–MS instruments and are thus available at a much higher temporal frequency (2 h). The data from ZEP and CGO were averaged to monthly means.

There is a calibration scale difference among AGAGE, NOAA, and NILU observations. However, Prinn et al. (2018) noted that the difference between the NOAA-2016 and SIO-05 scales is smaller than 1%, and material compatibility tests confirm negligible biases between samples collected in glass and stainless steel containers (Wu, L. et al., 2001). More detailed information on the observations is provided in Table 1.

Table 1. Introduction to the sampling station and sampling information.

| Station                                     | Latitude,<br>longitude | Altitude (m asl) | Instrument type                | Resolution | n Amount of data |  |
|---------------------------------------------|------------------------|------------------|--------------------------------|------------|------------------|--|
| Zhongshan, China<br>(ZOS)                   | 69.4° S, 76.4° E       | 71               | Steel canister                 | 2 w        | 25               |  |
| South Pole Observatory, United States (SPO) | 90° S, 24.8° W         | 2,841            | Glass flask and steel canister | 2 w        | 9–24             |  |
| Palmer, United<br>States (PSA)              | 64.8° S, 64.1° W       | 10               | Glass flask                    | 2 w        | 5–24             |  |
| Trollhaugen, Norway (TRL)                   | 72° S, 2.5° E          | 1,553            | Steel canister                 | 1 w        | 42–48            |  |
| Cape Grim,<br>Australia (CGO)               | 40.7° S, 144.7° E      | 94               | Glass flask and steel canister | 1-2 w      | 6–45             |  |
| Zeppelin, Norway<br>(ZEP)                   | 78.9° N, 11.9° W       | 475              | Online GC instrument           | 2 h        | 3,064–3,329      |  |

## 3 Methods


## 3.1 HYSPLIT model

Air mass backward trajectories were computed using the Hybrid Single-Particle Lagrangian Integrated Trajectory 175 (HYSPLIT) model (Stein et al., 2015). HYSPLIT is a complete system for computing simple air parcel trajectories, as well as complex transport, dispersion, chemical transformation, and deposition simulations. The model is widely used to analyze the transport and diffusion of atmospheric pollutants, gases, aerosols, and dust by calculating backward air quality trajectories at various altitudes and times using meteorological data (Ding, M. et al., 2020; Fan, S. et al., 2021; Chen, S. et al., 2023). The model calculation method is a hybrid between the Lagrangian approach and the Eulerian methodology. In the Lagrangian model, air concentrations are obtained by summing virtual air parcels of zero volume, which are advected

through grid cells along its trajectory (Escudero et al., 2006; Draxler and Hess, 1998). In the Eulerian model, air concentrations are calculated via the integration of mass fluxes in each grid cell on the basis of their diffusion, advection, and local processes. Gridded meteorological data were sourced from the National Centers for Environmental Prediction (NCEP), namely, from the Global Data Assimilation System (GDAS1, 1° horizontal resolution), which is operated by NOAA and provides 23 vertical levels, from 1,000 to 20 hPa (Draxler and Hess, 1998; Stein et al., 2015) (https://www.ready.noaa.gov/data/archives/gdas1/; last access: 13 April 2025). In this study, the TrajStat tool (MeteoInfoMap plugin for air mass trajectory statistics; Wang Y. (2014)) was used to drive the model.

Because most F-gases are long-lived gases (Supplementary Materials Table S1), their properties in the atmosphere remain relatively stable. Therefore, they can be transported with air masses across long distances. According to the above characteristics and model operation, the backtracking time was set to 720 h (30 days), and the temporal resolution is 1 h. The boundary layer top in the Antarctic region is low, so the above ground level (agl) was set to 500 m to represent the top of the boundary layer or the lower portion of the free atmosphere in the area. This height typically reflects the features of air mass movements from regional and long-range sources rather than the direct influence of local sources close to the surface. After calculating backward trajectories over 12 months of 2021, a combined backward trajectory for the entire year was obtained, yielding 8,760 trajectories for each station.

To analyze the composition of the backward trajectories of air masses at the various sites and their proportions, trajectory clustering analysis was performed. Trajectories with similar directions and rates were merged and classified to represent a class of air mass transport paths. Ward's least variance method was employed for clustering (Stein et al., 2015; Wang, S. et al., 2015). The basic idea is to classify trajectories into n classes and then reduce each class. It is necessary to satisfy the sum of the Euclidean distances of the pairs of trajectories and the minimum gap between the original distances. Ultimately, each trajectory is reduced to the appropriate class. The Euclidean distance can be calculated as follows:

$$d_{12} = \sqrt{\sum_{i=1}^{n} \left[ (X_{1i} - X_{2i})^2 + (Y_{1i} - Y_{2i})^2 \right]}$$
 (2)

where  $d_{12}$  is the Euclidean distance between traces 1 and 2;  $X_{1i}$  and  $Y_{1i}$ ,  $X_{2i}$  and  $Y_{2i}$  are the positions of point i on trajectory 1 and 2, respectively; and n is the number of trajectory points after 720 h.

To determine a reasonable number of clusters, the total spatial variance (TSV) method (Guan, Q. et al., 2019) is generally applied, and the TSV is used to obtain the number of clusters. There is high consistency in TSV-based selection among the four Antarctic stations, all of which exhibit 3 clusters.

## 3.2 PMF model





To better analyze the anthropogenic source of F-gases at ZOS, a positive matrix factorization (PMF) model was used for source analysis. On the basis of a matrix consisting of the concentrations of diverse chemical species, the objective of the

PMF model is to determine the number of F-gas source factors, the chemical composition profile of each factor, and the contribution of each factor to species (Paatero and Tapper, 1994). The PMF model aims to decompose the data matrix  $X_{(n\times m)}$  into  $G_{(n\times p)}$ ,  $F_{(p\times m)}$  and  $E_{(n\times m)}$  matrices. The matrix representation is as follows:

$$X_{(n\times m)} = G_{(n\times p)} \times F_{(p\times m)} + E_{(n\times m)}$$
(3)

where  $X_{(n\times m)}$  is the matrix of chemical concentrations measured at the receptor station;  $G_{(n\times p)}$  denotes the factor contribution matrix;  $F_{(p\times m)}$  is the factor spectrum matrix;  $E_{(n\times m)}$  is the residual matrix; and n is the number of samples. m chemical composition species; p is the number of factors (sources of pollution) resolved.

Element  $e_{ij}$  of the residual matrix  $E_{(n \times m)}$  is the residual value not explained by the model data value  $x_{ij}$ . Notably,  $x_{ij}$  can be calculated as follows:

$$x_{ij} = \sum_{k=1}^{p} g_{ik} f_{kj} + e_{ij}$$
 (4)

where  $x_{ij}$  is the concentration of the jth species in the ith sample;  $g_{ik}$  is the kth source for the ith sample;  $f_{kj}$  is the concentration of the jth element from the kth source; and  $e_{ij}$  is the residual concentration of the jth species in the ith sample.

The objective function Q is weighted via the least squares method, using residuals and uncertainty (Reff et al., 2007):

$$Q(E) = \sum_{i=1}^{n} \sum_{j=1}^{m} \left(\frac{e_{ij}}{u_{ij}}\right)^{2}$$
 (5)

where n and m are the numbers of samples and species, respectively, and u<sub>ij</sub> is the uncertainty determined by the detection limit for each species.

The uncertainty can be calculated as follows:

$$u_{ij} = \begin{cases} \frac{5}{6} \times MDL_j, C \le MDL \\ \sqrt{(MU \times C_{ij})^2 + (0.5 \times MDL_j)^2}, C > MDL \end{cases}$$
(6)

where MU is the error fraction, usually between 5% and 20%, which is 10% in this study(Wu, Z et al., 2025); C<sub>ij</sub> is the concentration of the jth species in the ith sample; and MDL<sub>j</sub> is the minimum detection limit for the jth species. The measured concentrations of F-gases in this paper are all greater than the MDL of the instrument.

The PMF model is supported by U.S. EPA PMF software version 5.0. The specific process includes data preparation, data input and preliminary inspection, basic calculation, rotation calculation, source analysis result calculation, and assessment of the results.

## 235 4 Results and discussion

## 4.1 Concentration




#### 4.1.1 Annual concentration

The annual concentrations and standard deviations across the four Antarctic observation stations are listed in Table 2. With respect to the annual mean concentration, the concentrations of seven F-gases at the Zhongshan site were the highest. The maximum values of HFC-125 and HFC-143a occurred at the TRL site. For HFC-152a, the SPO site exhibited the highest annual average concentration among the four sites. The concentration of HFC-236fa at SPO was approximately 1.5 times that at other sites. It is speculated that it was affected by the impact of pollution. In terms of standard deviation, PSA exhibited the lowest standard deviation in terms of the annual concentration, likely due to its reporting of monthly mean concentrations, which smooths out day-to-day variations. The average standard deviations for each gas at the ZOS and TRL stations (0.7 and 0.75, respectively) were greater than those at the SPO and PSA stations (0.58 and 0.52, respectively). It is hypothesized that ZOS and TRL may be more frequently disturbed by polluted air masses.

The coefficient of variation (CV; Table 2) of the annual mean concentration at these four stations was calculated. The mean CV value was 3.16%(except HFC-236fa), which is below the 5% threshold suggested by Jalilibal et al. (2021). Consequently, the data from ZOS can be considered representative of the background atmospheric composition in Antarctica, especially in East Antarctica.

Table 2. Annual ambient air concentrations of F-gases at the four Antarctic stations in 2021.

| F-gases         | ZOS              | PSA                                                  | SPO                                                  | TRL                                                  | Mean  | Standard deviation (SD) | CV (%) |
|-----------------|------------------|------------------------------------------------------|------------------------------------------------------|------------------------------------------------------|-------|-------------------------|--------|
| HFC-134a        | $112.9 \pm 2.3$  | $112.4 \pm 2.0$                                      | $112.6 \pm 2.1$                                      | $111.8 \pm 2.2$                                      | 112.4 | 0.5                     | 0.4    |
| HFC-23          | $34.8\pm\!0.6$   | -                                                    | -                                                    | $34.0 \pm\! 0.7$                                     | 34.4  | 0.6                     | 1.7    |
| HFC-125         | $34.4\pm\!1.1$   | $32.6\pm\!0.9$                                       | $32.7 \pm 1.0$                                       | $34.5\pm\!1.4$                                       | 33.6  | 1.0                     | 3.0    |
| HFC-143a        | $26.5\pm\!0.7$   | $25.3 \pm\! 0.4$                                     | $25.6 \pm \hspace{-0.05cm} \pm \hspace{-0.05cm} 0.6$ | $26.6 \pm \hspace{-0.05cm} \pm \hspace{-0.05cm} 0.8$ | 26.0  | 0.7                     | 2.6    |
| HFC-32          | $24.5\pm\!1.4$   | $20.6\pm\!1.0$                                       | $21.0\pm\!1.2$                                       | $24.1 \pm 1.4$                                       | 22.5  | 2.0                     | 8.8    |
| HFC-152a        | $4.8 \pm\! 0.3$  | $4.9\pm\!0.3$                                        | $5.0\pm\!0.3$                                        | $4.8\pm\!0.3$                                        | 4.8   | 0.1                     | 1.7    |
| HFC-227ea       | $1.9 \pm\! 0.1$  | $1.7\pm\!0.1$                                        | $1.7\pm\!0.1$                                        | $1.9\pm\!0.1$                                        | 1.8   | 0.1                     | 4.3    |
| HFC-236fa       | $0.22 \pm\! 0.0$ | $0.2\pm\!0.0$                                        | $0.35\pm\!0.01$                                      | $0.21~{\pm}0.0$                                      | 0.25  | 0.1                     | 28.8   |
| PFC-116         | $5.3 \pm \! 0.2$ | $4.9\pm\!0.1$                                        | $4.9 \pm\! 0.1$                                      | -                                                    | 5.0   | 0.2                     | 4.9    |
| $CF_4$          | $86.7 \pm 0.9$   | $86.2 \pm \hspace{-0.07cm} \pm \hspace{-0.07cm} 0.3$ | $86.3 \pm 0.4$                                       | $86.3 \pm \hspace{-0.07cm} \pm \hspace{-0.07cm} 0.3$ | 86.4  | 0.2                     | 0.3    |
| NF <sub>3</sub> | $2.6\pm\!0.1$    | $2.3 \pm 0.1$                                        | $2.3 \pm 0.1$                                        | -                                                    | 2.4   | 0.2                     | 7.1    |

## 4.1.2 Monthly and discrete daily concentration

In order to explore regional differences of F-gases, we calculated the monthly average concentration of the sampled concentration data and compare it with the stations in the Arctic and mid-latitudes of the Southern Hemisphere (Figure 2). There was a rapid increase in the concentrations of HFC-134a, HFC-125, HFC-143a, HFC-32, and HFC-227ea at the ZEP station toward the end of 2021. This increase of concentration in 2021 underscores the potential impact of regional emissions on the atmospheric composition at this location, highlighting the importance of continuous monitoring for understanding the dynamics of these compounds in different environments. ZEP, which is closer to human activities, exhibited higher concentrations than the Antarctic stations and the midlatitude CGO station in the Southern Hemisphere. The concentrations measured at CGO were very close to those at SPO and PSA. This demonstrates that, in the Antarctic, the relatively high concentrations at TRL and ZOS may be caused by local pollution or special air mass transport. However, the four Antarctic stations exhibited slow increases or even slight decreases. Previous study (Xiang, B et al., 2014; Lakshmanan, S. et al., 2023; Annadate, S. et al., 2025) revealed pronounced seasonal variations in the global emissions of HFC-134a, a major refrigerant, with summer emissions two to three times higher than those in winter.

In terms of seasonal differences of variation, HFC-152a concentration exhibited opposite patterns between the Northern and Southern Hemispheres. Given its relatively short atmospheric lifetime of 1.5 years, which is influenced by both emissions and atmospheric sinks, the variation in the HFC-152a concentration did not exhibit an obvious upward. However, the factors that drive the change in the HFC-152a concentration are not clear, nor is it clear whether mainly sinks or emissions are responsible for seasonal variations in the HFC-152a concentration. HFC-236fa exhibited satisfactory mixing globally and confirmed the presence of local contamination at SPO. The PFC-116 concentration at ZOS was even greater than that at ZEP, which should be confirmed through further observation.

Figure 2: Monthly concentrations of F-gases at the six stations in 2021.



We used unaveraged discrete daily concentrations to make linear fitting. In addition to the monthly data, discrete daily concentrations(Figure 3) were measured at ZOS, the slope of the line obtained through linear fitting is positive for several F-gases, including HFC-134a, HFC-23, HFC-125, HFC-143a, HFC-32, HFC-227ea, and NF<sub>3</sub> (p<0.05). Among these, HFC-134a exhibited the most notable increase, with a concentration rise of 5-6 ppt during 2021. HFC-32 showed the highest annual growth rate of 13.7%. Conversely, gases such as CF4, HFC-152a, HFC-236fa, and PFC-116 showed no significant increase or decrease. For HFC-32, the concentration increase at ZOS and TRL in 2021 was approximately 1 ppt, and the increase from October to December was obvious compared with that during the first half of the year. The HFC-236fa, PFC-116, and NF<sub>3</sub> concentrations remained almost unchanged or slightly increased in 2021. The increase or decrease of HFC-152a is seasonal, and the HFC-152a concentration was greater at SPO. HFC-236fa attained the maximum concentration increase in 2021, reaching up to twice its initial level. For HFC-32, HFC-143, HFC-32, and HFC-227ea, the concentrations

at ZOS and TRL were generally higher than those at PSA and SPO. For PFC-116 and NF<sub>3</sub>, which were not measured at TRL, the concentration at ZOS was also higher than those at SPO and PSA. Both ZOS and TRL are located in the East Antarctic coastal area, and their similar geographic locations may be the reason for the high concentration synchronization. Therefore, the input air mass trajectory at each station was calculated and analyzed to explore the causes of the observed concentration differences (Section 4.2). In addition, the errors associated with the sample and analysis system and the calibration scale may be partially explained. At ZOS, the concentrations of five types of F-gases (HFC-134a, HFC-23, HFC-32, PFC-116, and CF<sub>4</sub>) increased sharply in May and decreased in June. All the data generated through instrument processing of the canister samples were normal. In contrast, the concentrations of the other F-gases remained within the normal fluctuation range. Therefore, the pollution source carrying the specific F-gases can be attributed to the input.

Figure 3: Discrete daily concentrations of F-gases at the four Antarctic stations.


## 4.2 Effects of air mass transport







Backward trajectory analysis via the HYSPLIT model revealed distinct atmospheric transport patterns to the four Antarctic stations (Figure 4). On the basis of cluster analysis, the air masses arriving at ZOS were categorized into three dominant pathways, with relative contributions of 51.83%, 37.14%, and 11.03%. All the clusters originated from the ocean 30 days prior, forming an approximately circular path due to zonal westerly wind circulation. The primary pathway (88.97%) exhibited landfall over Princess Elizabeth Land along the Indian Ocean coast before it reached ZOS via westerly advection. Only 11.03% of the air masses made landfall over East Antarctic at 145°E, moving into Wilkes Land before migrating east of ZOS. Vertical profiling revealed that air parcels initiated at ~750 hPa over oceanic regions experienced orographic uplift followed by katabatic acceleration during eastern ingress into the ZOS area.

TRL exhibited comparable trajectory clustering (46.51%, 36.03%, and 17.46%, respectively) with similar recirculation characteristics, although it was distinguished by landfall dynamics. The dominant cluster (46.51%) made primary contact with the Antarctic continent in the Atlantic sector (30° E), whereas the secondary cluster (36.03%) originated from the southern Indian Ocean coast of Enderby Land. Notably, the tertiary trajectory (17.46%) encompassed a complex transcontinental loop: after initial landfall in Oates Land, it traversed Wilkes Land, re-entered the Indian Ocean, and completed a circumpolar transit before reaching TRL. The vertical displacement patterns revealed gradual subsidence interrupted by intermittent orographic lifting events.

In contrast to ZOS and TRL, SPO demonstrated bifurcated oceanic influences with cluster proportions of 49.03% (Pacific–Atlantic–SPO), 34.23% (Indian Ocean–Oates Land–SPO), and 16.75% (30°E–Antarctic Peninsula–SPO). All trajectories maintained marine origins but diverged in terms of hemispheric routing—approximately half originated from Pacific sectors versus Indian Ocean sources. Vertical development initiated from 700–750 hPa, with notable uplift to 600 hPa through terrain forcing before final subsidence over SPO.

The clustering configurations (40.43%, 32.89%, and 26.68%) at PSA revealed an exclusive Indian Ocean provenance at 750 hPa. Westerly circulation drove progressive subsidence along a west–east axis across the Pacific before ingress into the PSA area. This marine-dominated transport regime, characterized by elevated humidity and enhanced pollutant dispersion capacity levels (Zheng, X et al., 2023; Wu, Z. et al., 2025), is correlated with the observed reduced contaminant concentrations at PSA relative to those at ZOS and TRL. This partly explains the lower concentration at PSA than those at ZOS and TRL.

While backward trajectory analysis can reveal synoptic transport mechanisms, it is not suitable for identifying specific emission sources. All 30-day backward trajectories remained embedded within the Antarctic circumpolar circulation, precluding clear terrestrial source attribution. This suggests that a more detailed atmospheric chemical transport model should be employed to trace and quantify emission sources, as this can provide a clearer understanding of the emission flux and its transport dynamics in the region. Cluster analysis revealed that the concentrations of F-gases from the three air

masses transmitted to ZOS were consistent (Table S4), as were those at TRL, SPO, and PSA. This suggests that the atmospheric background feature of uniform Antarctic mixing may be observed at all four of the Antarctic locations included here.

Figure 4: Three clusters of HYSPLIT-derived backward trajectories at the four Antarctic stations in 2021.

## 4.3 Source apportionment analysis at ZOS

The potential sources of F-gases and their relative contributions to each category were determined via the PMF model. HFC-236fa, CF4, and PFC-116 were not included because of their low correlation coefficients during concentration fitting, which could affect the accuracy of the model. F-gases were derived from a variety of sources, as detailed in Table 3. Four isolation factors were extracted according to the composition profiles depicted in Figure 5. The percentages of factors 1 to 4 were 24.69%, 23.28%, 29.95%, and 22.08%. Residual analysis of the PMF results was conducted, and the results are shown in Supplementary Materials Figure S3. The histogram of the frequency distribution of the residuals exhibits a symmetrical bell shape. The mean value is close to 0 (0.00079), indicating that there is no significant deviation in the fit of the model to the observed data. The residuals are mainly distributed within the (-0.5~0.5) range, and the standard deviation is σ = 0.12, indicating that the fitting error is small and concentrated. The Gaussian fitting R² value is 0.99323, indicating that the residual distribution is highly consistent with the normal distribution.

Table 3. Main applications of 8 types of F-gases

| Species   | Application a                                     |  |  |
|-----------|---------------------------------------------------|--|--|
| HFC-134a  | Refrigerant, foaming agent, dry etching agent     |  |  |
| HFC-143a  | Refrigerant                                       |  |  |
| HFC-23    | Fire extinguisher, dry etching agent, refrigerant |  |  |
| HFC-32    | Refrigerant, dry etching agent                    |  |  |
| HFC-125   | Refrigerant, fire extinguisher, dry etching agent |  |  |
| HFC-152a  | Foaming agent, aerosol inhale, refrigerant        |  |  |
| HFC-227ea | Fire extinguisher, aerosol inhaler                |  |  |
| $NF_3$    | Dry etching agent                                 |  |  |

a: Collected from multiple studies (O'Doherty et al., 2004, 2014; Velders et al., 2009; Montzka et al., 2015, 2019; M. Takeda et al., 2021; Vollmer et al., 2011; UNEP, 2022; Kim et al., 2014, 2021; Arnold et al., 2018).


Figure 5: Plots of the factor profiles and individual contributions of each factor obtained via the PMF model.

In factor 1, HFC-227ea and HFC-23 exhibited relatively high loadings (53.59% and 38.02%, respectively). HFC-227ea is a fire retardant that has replaced halon-1301 and is also used as a propellant in metered-dose inhalers (MDIs) (UNEP, 2022; Vollmer et al., 2011). HFC-23 is used in halon-1301 (CF<sub>3</sub>Br) production, very low-temperature refrigeration, and specialty fire extinguishers (Oram et al., 1998; Miller et al., 2010, Simmonds et al., 2018), with very low emissions from deliberate use (Takeda et al., 2021). Hence, HFC-23 has mainly been discharged from HCFC-22 production plants into the atmosphere (Montzka et al., 2019). Therefore, factor 1 can be assigned as a potential source of fire extinguishing agents and industrial byproducts.

In factor 2, HFC-32 and HFC-152a attained relatively high loadings (36.7% and 32.17%, respectively). HFC-32 is a replacement for HCFC-22 and is applied mainly in the fields of air conditioning and refrigeration. For example, the refrigerant blends R-410A (50% HFC-32 and 50% HFC-125 by weight) and R-407C (23% HFC-32, 52% HFC-134a and 25% HFC-125 by weight) are azeotropes used in stationary air-conditioning systems (Velders et al., 2022). HFC-152a is used mainly as a foaming agent in aerosol inhalers. Therefore, factor 2 can be assigned as a potential source of refrigeration and foaming agents.

In factor 3, HFC-23 and HFC-134a exhibited relatively high loadings (44.61% and 34.99%, respectively). HFC-23 exhibits the second-highest radiative forcing among all HFCs and fluorinated gases just after HFC-134a (14.3 mWm<sup>-2</sup>) (Montzka et al., 2019). HFC-143a is used mostly as a working fluid component in refrigerant blends R404A (52% HFC-143a, 44% HFC-125 and 4% HFC-134a by weight) and R-507A (an azeotropic blend of 50% HFC-125 and 50% HFC-143a) for low- and medium-temperature commercial refrigeration systems, thereby replacing R-502 (a blend containing HCFC-22 and CFC-115) (Velders et al., 2009). Other minor uses are R-408A, R427A, and R-428A as replacements for R-502 and HCFC-22. HFC-134a is the preferred substitute for CFC-12, primarily in refrigeration and air-conditioning applications (O'Doherty et al., 2014). Therefore, factor 3 can be assigned as a potential source of industrial byproducts and refrigeration gases.

In factor 4, HFC-152a, HFC-32, and NF<sub>3</sub> attained relatively high loadings (67.82%, 63.29%, and 40.02%, respectively). NF<sub>3</sub> has been used in the semiconductor industry and in the production of photovoltaic cells and flat-panel displays as an etching agent (Arnold et al., 2018). Therefore, factor 4 can be assigned as a potential source of foaming, refrigeration, and electronics products.

# 5 Data availability




All F-gas data from ZOS presented in this paper are archived and publicly accessible at the National Tibetan Plateau Data Center via the permanent DOI: https://doi.org/10.11888/Atmos.tpdc.302283 (Tian, B., et al., 2025). The dataset is named "Near-surface fluorinated greenhouse gas observations from the Chinese Antarctic Zhongshan Station expedition in 2021". The dataset includes the near-surface atmospheric concentrations of 11 F-gases measured at ZOS, Antarctica, in 2021. The F-gas species are HFC-134a, HFC-23, HFC-125, HFC-143a, HFC-32, HFC-152a, HFC-227ea, HFC-236fa, PFC-116, CF<sub>4</sub>, and NF<sub>3</sub>.

This dataset contains gas concentration information for 25 samples, with each row corresponding to one sample. The first column represents the sampling time (in YYYY-MM-DD format), and the remaining columns indicate the concentration values of the various fluorinated gases, in units of ppb. The data were collected from January 2021 to December 2021, with a sampling frequency of once every two weeks. All the data passed the  $3\sigma$  test. Users are advised to cite both this publication and the dataset DOI when using the data. Additional supporting information is available upon request from the corresponding authors.

#### 6 Summary

This study provides a comprehensive analysis of the spatiotemporal characteristics and source apportionment of F-gases at 390 ZOS and three other Antarctic stations in 2021. The F-gas concentrations across the four stations—ZOS, TRL, SPO, and

PSA—exhibited distinct temporal variations and spatial distribution patterns. The concentrations of most of the target F-gases showed significant growth characteristics. ZOS and TRL exhibited higher mean F-gas concentrations with greater fluctuations, whereas PSA and SPO demonstrated lower annual mean concentrations with lower fluctuations. These findings highlight the importance of continuous monitoring at these sites to better understand regional and global atmospheric dynamics.

Via the use of the HYSPLIT model for backward trajectory clustering analysis, we found that the contributions of different trajectory clusters were nearly identical across all four stations. This indicates a well-mixed distribution of F-gases over Antarctica. The minimal concentration differences between trajectories validate the suitability of these stations as background observation points. Specifically, ZOS, situated close to the Antarctic ice sheet and less notably affected by human activities, effectively represents the background atmospheric conditions in East Antarctica. The data uniformity underscores the reliability of these stations for studying global climate change and the atmospheric composition.

The PMF model revealed four major contributing sources, namely, industrial byproducts and firefighting agents (factor 1. 24.69%), refrigeration and foaming agents (factor 2, 23.28%), industrial refrigeration agents (factor 3, 29.95%), and electronics manufacturing (factor 4, 22.08%). Overlaps were observed among the identified source factors, indicating that some F-gases may originate from multiple sources or share similar atmospheric transport pathways and that F-gases in the Antarctic are affected by the compounding effect of global emissions from multiple industries.

There are some limitations about the potential influence of polar singularities on trajectory simulations and clustering analysis. In addition, based on the current time resolution and uncertainty of our data, it is difficult to research the impact of polar day and night on F-gases. Future studies should aim to combine long-term continuous observations with high-resolution chemical transport modeling to quantify the proportions and variations in the contributions of different emission sources and to assess the actual regulatory effects of the MP and its amendments on the Antarctic atmospheric composition.

## **Author contribution**





BT, RN and BY designed the experiments and wrote the manuscript; XL carried out the experiments; BTand RN analyzed the experimental results; MD, BT, and BY revised the manuscript. YZ, DZ, WS, CL, XW, JT and MD discussed the results.

#### **Competing interests**

The authors declare that they have no con-flict of interest.

## Financial support

This research has been supported by the National Natural Science Foundation of China (grant no. 42201151) and the Basie 420 Fund of the Chinese Academy of Meteorological Sciences (grant nos.2023Z004 and 2024Z007).

#### Acknowledgements

This work was supported by the National Natural Science Foundation of China (42201151) and the Basic Research Fund of the Chinese Academy of Meteorological Sciences (2023Z004, 2024Z007). The observation were carried out by during the Chinese National Antarctic Research Expedition (CHINARE) at the Zhongshan Station. We are also grateful to NOAA for providing the HYSPLIT model and GFS meteorological files. Many thanks to the relevant scholars of the NOAA/HATS project and the EBAS/NILU database for data support.

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
