# Peer review of "Concentration changes of atmospheric F-gases and analysis of their potential sources at Zhongshan Station, Antarctica, 2021"

_Earth System Science Data, 2025_

## Author Comment (AC2)

**Reply for Anonymous Referee #2**

**Nan et al. report a one-year high-quality measurement of 11 fluorinated gases (F-gases) at Zhongshan Station in Antarctica, providing valuable insights into F-gas concentrations over East Antarctica. The paper is well-written overall, but the clarity and rigor of the modeling and data analysis sections (particularly Section 3 and Section 4) could be enhanced. Below are specific suggestions for improvement.**

1. L193-197: The HYSPLIT model should be introduced (e.g., as a trajectory model) before discussing Eulerian and Lagrangian frameworks. This would improve readability for non-specialists.

**Reply:** Thank you to the reviewers for your valuable suggestions. In the revised draft, before introducing the Euler framework and the Lagrange framework, we added a background description of HYSPLIT as a backward trajectory model. The revised paragraph is as follows:

Air mass backward trajectories were computed using the Hybrid Single-Particle Lagrangian Integrated Trajectory (HYSPLIT) model (Stein et al., 2015). HYSPLIT is a complete system for computing simple air parcel trajectories, as well as complex transport, dispersion, chemical transformation, and deposition simulations. The model is widely used to analyze the transport and diffusion of atmospheric pollutants, gases, aerosols, and dust by calculating backward air quality trajectories at various altitudes and times using meteorological data (Ding, M. et al., 2020; Fan, S. et al., 2021; Chen, S. et al., 2023). The model calculation method is a hybrid between the Lagrangian approach and the Eulerian methodology. In the Lagrangian model, air concentrations are obtained by summing virtual air parcels of zero volume, which are advected through grid cells along its trajectory (Escudero et al., 2006; Draxler and Hess, 1998). In the Eulerian model, air concentrations are calculated via the integration of mass fluxes in each grid cell on the basis of their diffusion, advection, and local processes. Gridded meteorological data were sourced from the National Centers for Environmental Prediction (NCEP), namely, from the Global Data Assimilation System (GDAS1, 1° horizontal resolution), which is operated by NOAA and provides 23 vertical levels, from 1,000 to 20 hPa (Draxler and Hess, 1998; Stein et al., 2015) (https://www.ready.noaa.gov/data/archives/gdas1/; last access: 13 April 2025). In this study, the TrajStat tool (MeteoInfoMap plugin for air mass trajectory statistics; Wang Y. (2014)) was used to drive the model.

2. L211: "classify n trajectories into n classes"? Doesn't make sense.

**Reply:** Sorry, The correct sentence this time should be: The basic idea is to classify trajectories into n classes and then reduce each class.

3. Eq. (2): The correct notation should be $X_{1i} - X_{2i}$ (difference between two measurements).

**Reply:** Thank you very much for your seriousness. I apologize for my carelessness. It has been modified.

4. L215: The terms "node" and "track" are not clearly defined, and are probably unnecessary to use. Please avoid excessive jargon and provide clear definitions for technical terminology.

**Reply:** The terms have already been replaced in the paper. The replaced terms do not change their original meanings and are more understandable. The revised sentence is as follows:

where $d_{12}$ is the Euclidean distance between traces 1 and 2; $X_{1i}$ and $Y_{1i}$, $X_{2i}$ and $Y_{2i}$ are the positions of point i on trajectory 1 and 2, respectively; and n is the number of trajectory nodes after 720 h.

5. Section 3.2. Eq (3): The values for n, p, and *m* should be explicitly stated.

Reply: We are very sorry that we missed the introduction of the meanings of these subscripts. In the text, I supplemented the meanings of these three letters. The following is the specific content:

n represents the number of samples;

m denotes the number of chemical species;

p is the number of factors resolved by the analysis (typically corresponding to distinct pollution sources or source categories).

6. Eq(6): The uncertainty appears to depend not only on the detection limit but also on $C_{ij}$ and MU. The statement above is inaccurate. In addition, justify the choice of *MU = 10%*—is this based on prior literature or empirical validation? Is it justifiable to use a single value for different species? Also, what fraction of the data actually falls below the MDL? This information should be provided.

**Reply:** Thank you very much for your constructive suggestions. We have revised the manuscript to clarify the sources of measurement uncertainty. Measurement uncertenty is the error fraction, usually between 5% and 20%, which is 10% in this study. "MU = 10%" is this based on prior literature(Wu, Z et al., 2025). The methods he used and our used were the same. Specifically, the sampling process was tank sampling combined with laboratory analysis, and the analytical methods were both the pre-concentrator system combined with the GC-MS method.

The measured concentrations of F-gases in this paper are all greater than the MDL of the instrument. Therefore, the formula mainly uses the one when C>MDL. However, in order to present the calculation method of uncertainty completely, the formula under the condition of C≤MDL was also recorded. Sorry for causing you trouble. I have also marked the actual usage of this formula in the article.

$$u_{ij}=\begin{cases} \dfrac{5}{6}\times MDL_j, & C\leq MDL \\ \sqrt{(MU\times C_{ij})^2+(0.5\times MDL_j)^2}, & C>MDL \end{cases}$$

7. Currently, Sections 4.1.2 (monthly trends) and 4.1.3 (daily trends) present overlapping information. Merging them into a single, more coherent discussion would improve readability. In addition, the discussion should be better structured around Zhongshan Station (ZOS) rather than beginning the discussion with ZEP Station, which is confusing.

**Reply:** Thank you very much for your suggestions about writing logic. The section "Discrete daily concentration" and section "Monthly concentrations" have been merged, and the recurring conclusions have been removed. The combined chapters discuss the spatial differences, seasonal differences and changing trends of F-gases concentration in three aspects. The discussion part highlighted the importance of Zhongshan Station.

8. L276: It is ambiguous from the context whether the discussion refers to HFC-134a alone or multiple HFCs (HFC-125, HFC-143a, HFC-32, HFC-227ea). Please clarify.

**Reply:** Thank you for your careful suggestions. Specifically, the previous seasonal emission differences are only the analysis and reference of HFC-134a( Xiang, B et al., 2014;Annadate, S. et al., 2025; Lakshmanan, S. et al., 2023), while the research of emission variation of remaining F-gases was limited. But the OH in the following text is the common convergence of multiple HFCs. Because the reaction with OH is the main sink (>90%) for multiple HFCs (Thompson et al., 2024).

9. L277-284: The attribution of concentration differences to heterogeneous OH production lacks sufficient evidence. The general understanding is that OH oxidation is most active in the tropics, not locally in Antarctica, through heterogeneous OH production. Additional analysis or references should support this claim—otherwise, I suggest removing this speculative statement.

**Reply:** Thank you for your comments. I have removed this speculative statement. Referring to previous studies(Kozlov, S. N. et al., 2003; Thompson et al., 2024), HFCs are not appreciably photolyzed by solar radiation in the visible and near UV portion of the spectrum; therefore, their residence times in the Earth's atmosphere are primarily controlled by reactions with the OH radical in the troposphere. But I haven't found any articles explaining the OH radical mechanism about HFCs in the Antarctic region. So I'm not sure about the applicability of the above speculation in Antarctica, so I deleted this inference.

10. In 327: over East China?
**Reply:** Sorry, that should be "East Antarctic". It has been modified.